# Frequency-Dependent Amplification of Head Motion in Infant Rockers: A Segmental IMU-Based Signal Analysis

**DOI:** 10.3390/jcm14238301

**Published:** 2025-11-22

**Authors:** Alina Głowińska, Sebastian Glowinski

**Affiliations:** 1Independent Researcher, 74-024 Koszalin, Poland; alina.glowinska@o2.pl; 2Institute of Health Sciences, Pomeranian University in Slupsk, Westerplatte 64, 76-200 Slupsk, Poland; 3Institute of Physical Culture, The State Academy of Applied Sciences in Koszalin, Leśna 1, 75-582 Koszalin, Poland

**Keywords:** infant rockers, inertial measurement units (IMUs), frequency analysis, wavelet transform

## Abstract

**Highlights:**

**What are the main findings?**

Head motion in infant rockers demonstrates strong frequency-dependent amplification, with the highest angular displacements and peak accelerations occurring in the head segment.The abdomen acts as a damping transition zone, while the gluteal region remains almost static, serving as the mechanical input.A distinct amplification of motion energy is observed in the 10–12 Hz range, indicating inertial and potentially resonant effects.

**What is the implication of the main finding?**

Even seemingly gentle rocking can lead to significant dynamic loading at the head due to upward frequency-dependent amplification.These results highlight the need to consider segmental dynamics and resonance effects when designing infant rockers.The findings have ergonomic and safety implications, especially for younger infants with more vulnerable neuromuscular control.

**Abstract:**

(1) **Background:** Passive rocking is commonly used to calm infants, yet its biomechanical impact on different body segments—particularly dynamic loading of the head and torso—remains insufficiently understood. (2) **Methods:** An infant doll was instrumented with IMUs placed on the head, abdomen, and gluteal region and subjected to controlled rocking in a standard infant rocker. Segmental responses were quantified using angular displacement, velocity, and acceleration, along with time–frequency analyses including wavelet transforms and inter-segmental transfer functions. (3) **Results:** The head showed the highest angular displacements and peak accelerations, predominantly in the sagittal plane, with pronounced oscillations in the 10–12 Hz range. The abdomen acted as a transitional damping zone, while the gluteal region remained largely static, serving as the mechanical input point. Frequency-domain results revealed upward amplification of motion energy, suggesting inertial and potentially resonant effects that intensify loads at the head. (4) **Conclusions:** Although base-level motion appears gentle, signal-level analysis reveals significant dynamic amplification toward the head. These findings underscore the importance of considering frequency-dependent transmission and segmental dynamics in infant rocker design, with implications for ergonomic safety, particularly in the early stages of development.

## 1. Introduction

Infant rockers are widely used in childcare as a convenient way to soothe infants through gentle, rhythmic motion. These devices are designed to mimic the calming effect of rocking, often helping babies fall asleep more easily [1]. Despite their popularity, there is a surprising lack of biomechanical research into how these motions interact with the infant’s developing body—especially in terms of mechanical forces transmitted to sensitive regions like the head and neck [2,3].

Infants in their first months of life are especially vulnerable to mechanical loading due to the immaturity of both their musculoskeletal and neurological systems. The infant brain, enclosed within a soft, flexible skull, is particularly sensitive to sudden accelerations and decelerations [4]. While infant rockers typically generate gentle movements, even low-magnitude, repetitive accelerations—especially in the anterior–posterior direction—may introduce mechanical forces that exceed what developing tissues are prepared to absorb [5,6]. The infant’s relatively large head mass, combined with weak neck musculature, makes the head especially susceptible to translational and rotational accelerations during motion [7,8].

Current safety standards for infant rockers primarily emphasize structural stability and restraint systems [9]. However, they often overlook key biomechanical parameters such as the magnitude and frequency characteristics of motion transmitted to the infant’s body [10]. Although some guidelines have been proposed to ensure that rocking remains within physiologically safe limits, they are not consistently reflected in commercial products or regulatory frameworks [11].

Existing research in this area tends to focus on high-level metrics such as overall device acceleration or general infant responses to rocking [12,13] rather than the precise distribution of motion across different body segments. Yet this information is critical—particularly since the mechanical dynamics experienced by the head can differ substantially from those of the torso or pelvis. Rotational accelerations, in particular, have been identified as more damaging to neural tissue than linear forces alone, with the potential to induce shear and tensile strains associated with mild traumatic brain injury [14,15].

Despite these risks, studies directly examining how movement is transferred to different parts of the infant body during passive rocking remain limited [16,17]. This lack of data represents a significant gap in our understanding of how infant rockers interact biomechanically with the body—particularly in dynamic, frequency-dependent terms. Furthermore, while inertial measurement units (IMUs) have become increasingly popular in pediatric motion analysis [18,19], they have mostly been used to study active motor development rather than passive motion in devices like rockers.

To address this gap, the present study investigates how motion is transmitted to different infant body segments during passive rocking, with a particular focus on the head. We employed a life-sized anthropomorphic infant mannequin replicating the mass distribution and anatomical proportions of a 2–3-month-old infant. The mannequin was securely fastened into a commercially available rocker and manually rocked in the anterior–posterior direction at a consistent, high speed. IMUs were placed on the forehead, abdomen, and pelvis to capture three-dimensional angular velocity and orientation throughout the rocking cycle.

Motion data were collected at 100 Hz to capture high-resolution segmental kinematics. The signals were then analyzed using wavelet transformation and transfer function methods, allowing us to examine not only the amplitude of motion but also its dominant frequencies and transient dynamics [20]. This approach enabled a more precise understanding of how energy is transmitted through the body and whether certain segments—particularly the head—experience frequency-dependent amplification of motion.

By combining high-fidelity sensor data with frequency-domain signal analysis, this study provides novel insight into the biomechanical effects of infant rocking. Our results contribute to a growing body of work advocating for evidence-based safety assessments of infant products and may inform improved design standards aimed at minimizing mechanical risk while preserving the calming benefits of rocking.

The objectives of this study were as follows:Quantify segmental motion patterns during passive rocking using IMUs;Compare dynamic responses across head, abdominal, and gluteal regions;Analyze frequency-specific energy transmission using wavelet and transfer function analysis;Explore implications for rocker safety and infant biomechanics.

## 2. Materials and Methods

### 2.1. Experimental Measurement Setup

The study consisted of three main stages: the experiment, data processing, and wavelet analysis. The experiment was carried out using a life-sized infant dummy designed to replicate the body proportions and weight distribution of a 2–3-month-old baby. The dummy was placed in a commercially available infant rocker, which was then set into motion by a manually applied, rhythmic anterior–posterior (front-to-back) rocking movement (Figure 1a).

To capture the kinematic response of various body segments, three inertial measurement units (IMUs) were strategically positioned: one on the forehead (representing the head), one on the abdominal region (approximating the torso’s center of mass), and one beneath the gluteal area (attached to the rocker surface, under the pelvis). The IMUs were oriented consistently, with the X-axis aligned vertically (pointing upward along the body), the Y-axis pointing laterally to the left, and the *Z*-axis directed forward, perpendicular to the XY plane.

The measurement system was based on ProMove Mini sensors from Inertia Technology (AG Enschede, The Netherlands) [21]. The sampling frequency was set at 100 Hz, with angular velocity measured up to 2000°/s at a resolution of 0.007°/s. Each sensor was compact (51 × 46 × 15 mm) and lightweight (20 g, including battery). Data collection and synchronization were managed using the Inertia Gateway, serving as a central hub capable of synchronized acquisition with timing precision below 100 ns. Additionally, the Inertia Studio software (version 3.7.3.1) allowed real-time visualization of sensor data, over-the-air sensor reconfiguration, and adjustment of wireless parameters. All acquired data were logged for post-processing and analysis.

The simulation time was set to 30 s. The rocker was manually moved at the highest consistent speed possible on a flat, horizontal surface. The infant rocker used in the experiment weighed 3.05 kg, measured 43 cm in width (between the centers of the rocker rails), and was 63 cm in length. The seat was positioned 13 cm above the ground. The mannequin was securely strapped into the rocker using safety belts to prevent it from falling out.

After data collection, preliminary processing was carried out using (Figure 1b). Since the recorded signals exhibited periodic characteristics with potential variations in dominant periods over time, wavelet analysis was selected as an appropriate tool for further investigation.

At the outset, the radius of the arc along which the rocker moves were determined. The length of the rocker’s runner chord was measured (*c* = 0.52 m), as well as the distance from the midpoint of the chord to the arc—known as the sagitta (*h* = 0.03 m). The radius was then calculated using the following relation:(1)$$r=\frac{c^{2}}{8h}+\frac{h}{2}=\frac{0.2704}{0.34}+0.016=1.1417\;m$$

The larger the radius, the flatter the product, and vice versa. After determining the radius, the arc angle was calculated using the equation:(2)$$\alpha [rad]=2arcsin\left(\frac{c}{2r}\right)=2arcsin\left(\frac{0.52}{2\cdot 1.1417}\right)=2arcsin\left(0.2328\right)=2\cdot 0.2298=0.4595$$
or expressed in degrees(3)$$\alpha \left[deg\right]=\alpha \cdot \frac{180}{\pi }=26.32\;$$

The total range of motion of the rocker was approximately 26 degrees. It is important to note that the rocker is equipped with built-in stoppers at the ends of its curved rails to prevent excessive forward or backward tilt. When these stoppers are disengaged, the rocker can also be used as a stationary infant seat without the rocking function.

### 2.2. Wavelet Analysis Theory

In the context of biomechanics, motion signals are often non-stationary—meaning their frequency content changes over time. Traditional frequency analysis tools, such as the Fourier transform, while effective for steady-state signals, fall short when applied to dynamic movements like those observed during rocking. To address this, we incorporated continuous wavelet transform (CWT) into our analysis, offering a more suitable framework for investigating how motion energy evolves both in time and across frequency bands [22].

Wavelet analysis has been increasingly adopted in biomechanics due to its ability to capture localized changes in signal frequency, which is particularly relevant when studying the human (or infant surrogate) body in response to external motion [17]. In our study, the CWT provides a detailed view of how different body segments—namely the gluteal region, abdomen, and head—respond dynamically during passive rocking in an infant seat. This method allows us to pinpoint not only which frequency components dominate the movement, but also when they occur during the rocking cycle.

This is especially important when examining mechanical energy transmission through the body. For instance, understanding whether certain frequency ranges are amplified or dampened as they propagate upward from the pelvis to the head can offer valuable insight into potential risks to the infant’s developing musculoskeletal and neurological systems. Furthermore, the wavelet-based transfer function approach enables us to assess how motion introduced at the base (the rocker) is transferred through intermediate segments—something that simpler time-domain or frequency-domain analyses alone cannot fully resolve.

By applying wavelet methods, we are able to move beyond average values and observe the fine structure of motion responses, giving us a clearer understanding of the dynamic behavior of the infant surrogate across both time and frequency. This insight is crucial for evaluating the safety and mechanical impact of common soothing devices like infant rockers, especially in the context of early developmental stages when head control is limited and the brain is highly vulnerable.

For the wavelet analysis, we used the complex Morlet wavelet, a widely adopted choice in biomechanical and physiological signal processing [17]. The Morlet wavelet is particularly well-suited for analyzing motion data due to its optimal balance between time and frequency resolution, making it ideal for detecting localized oscillatory patterns in signals that change over time—such as those observed during rocking motion.

The Morlet wavelet consists of a complex sinusoid modulated by a Gaussian envelope, which allows it to capture both amplitude and phase information of specific frequency components. This property is especially valuable in our context, as we are not only interested in how much motion energy exists in a given frequency band, but also when and where it manifests during the rocking cycle. In addition, the Morlet wavelet is continuous and smoothly localized in time and frequency, making it a robust tool for identifying subtle dynamic features across body segments. Mathematical form of the complex Morlet wavelet can be expressed as [23]:(4)$$\psi \left(t\right)=\pi^{-1/4}\cdot e^{-j\omega_{0}t}\cdot e^{-t^{2}/2}$$

Each component of this expression plays a specific role:
$\psi \left(t\right)\;$—is the wavelet function in the time domain—a localized waveform used to analyze the structure of the signal at various scales and positions,$\pi^{-1/4}\;$—is a normalization constant that ensures the wavelet has unit energy. This allows for meaningful comparison of energy across different scales,$e^{-j\omega_{0}t}\;$—is the complex sinusoidal carrier—it defines the oscillatory behavior of the wavelet, centered at a frequency determined by the parameter $\omega_{0}$. This component allows the wavelet to extract frequency-specific information from the signal,$e^{-t^{2}/2}\;$—is the Gaussian envelope, which ensures that the wavelet is localized in time. This time localization is what gives the Morlet wavelet its ability to detect transient features and changing frequency content in non-stationary signals,$\omega_{0}\;$—is the central (dimensionless) frequency of the wavelet. A common choice is $\omega_{0}=6,\;$which provides a good balance between time and frequency resolution. Smaller values improve time localization at the expense of frequency resolution, and vice versa.

Together, these components form a wavelet that is well-localized in both time and frequency, making it an ideal tool for analyzing biomechanical signals—such as the accelerations observed in an infant surrogate during passive rocking. The ability to observe how energy at specific frequencies changes over time is what makes wavelet analysis, and the Morlet wavelet in particular, so powerful in the study of dynamic, non-stationary systems.

Given the quasi-periodic nature of the rocking motion—with shifting frequency content and damping characteristics over time—the Morlet wavelet allowed us to clearly visualize and track changes in signal structure from the gluteal region up to the head. This was essential in evaluating how specific frequency components were transmitted or amplified through the body, and whether the head experienced distinct dynamic behaviors compared to lower segments. In summary, the use of the Morlet wavelet in this study provided a high-resolution, time–frequency representation of the kinematic data, enabling a more nuanced understanding of motion propagation and segmental resonance patterns in the infant rocker system.

## 3. Results

### 3.1. Kinematic Analysis

The resulting data reveal important insights into how different parts of the infant’s body respond dynamically to passive rocking. The angular plots indicate that pitch was the dominant rotational component across all three sensor locations, which aligns with the direction of imposed motion. The head experienced the most pronounced angular displacements, with initial pitch angles reaching approximately −60 degrees, gradually decreasing in amplitude over the course of the recording (Figure 2). This damped oscillation pattern is indicative of a free, pendulum-like response, suggesting that the head, being elevated and less constrained than other body parts, undergoes the most dynamic movement during rocking. The roll and yaw components for the head remained relatively stable, which is expected given the unidirectional nature of the motion.

The abdominal region exhibited a similar oscillatory trend in pitch, though with reduced amplitude compared to the head. This suggests that while the abdominal area followed the rocking motion, its movement was partially dampened—likely due to the central positioning of the torso, greater contact area with the rocker surface, and possibly internal resistance from the doll’s body structure. Roll and yaw again remained minimal, reinforcing the observation that the motion was largely constrained to the sagittal plane.

In contrast, the gluteal region showed the lowest angular displacement. The pitch oscillations were still present but considerably more damped, while roll and yaw remained essentially flat. This result is consistent with the fact that this sensor was mounted on the rocker base and in direct contact with the supporting surface. As such, the gluteal segment acted as a mechanical reference, primarily transmitting motion rather than undergoing significant independent rotation.

Linear acceleration data complement the angular results and further highlight the differential dynamic response across body segments. The head showed the highest peak accelerations, reaching up to approximately 13 m/s^2^ shortly after the onset of rocking. These values gradually decreased over time, following a typical damped oscillatory pattern. This again suggests that the head is most susceptible to rapid changes in velocity due to its distance from the motion origin and reduced structural constraint. The abdominal segment recorded slightly lower acceleration peaks, yet the same damped oscillation trend was clearly visible, consistent with a moderately mobile but more stabilized segment. The gluteal acceleration data showed the lowest amplitudes and the most quickly attenuated response, reinforcing its role as the fixed or driven base.

Taken together, these results suggest a clear top-down gradient in mechanical response to anterior–posterior rocking, where the energy imparted at the base (rocker) is transmitted upward and gradually dissipated. The head, being furthest from the point of motion initiation and least constrained, exhibits the highest amplitude and most prolonged oscillatory behavior, potentially making it the most sensitive to prolonged or excessive movement. In contrast, the abdominal area behaves as a transitional zone, and the gluteal region remains relatively static. These findings have implications for the biomechanical safety of passive rocking in early infancy, particularly regarding head and neck dynamics. The data support the idea that while rocking may be gentle at the base, the resulting head movements can still be significant and should be considered in the design and usage recommendations for infant rockers, especially for very young or medically vulnerable infants.

In addition to the linear and angular displacement data, the analysis of angular velocity and angular acceleration (Figure 3) further illustrates how each body segment responded to the front-to-back oscillatory motion of the rocker. The top row of plots displays angular velocity in degrees per second for the forehead, abdomen, and gluteal regions. Across all three locations, the Y-axis component (ω_y_, blue line), which corresponds to rotation in the sagittal plane, is the dominant motion. This aligns with the expected behavior from anterior–posterior rocking, which induces primarily forward and backward rotation. Both the forehead and abdominal sensors recorded similar angular velocity profiles, with peak values of around ±70–80 deg/s, especially during the initial cycles of motion. These signals display clear sinusoidal waveforms that decay over time, consistent with a damped harmonic system. This again suggests that both the head and torso experienced substantial rotational motion in response to the input motion from the rocker, with the head showing slightly sharper peaks and longer-lasting oscillations. The gluteal region also shows rotation around the Y-axis, though with noticeably lower peak values and more damped oscillations, indicating that this area—being in direct contact with the base of the rocker—did not rotate as freely and likely served as the pivot or origin of movement for the upper body.

The bottom row of plots presents angular acceleration (in deg/s^2^), again showing that the sagittal plane (ε_y_) dominates across all body parts. The angular acceleration data is more sensitive to sudden changes in angular velocity, and this is evident in the presence of sharper, more discontinuous peaks—especially in the early cycles of movement. The forehead shows the highest angular acceleration spikes, reaching ±35 deg/s^2^, which reflects the rapid angular reversals experienced by the head as it swings back and forth. These large variations point to a significant dynamic load acting on the cervical region, which in real infants could be a point of concern, especially considering underdeveloped neck musculature at this age. The abdominal region again follows a similar pattern but with lower magnitude accelerations, peaking around ±20 deg/s^2^, indicating a smoother and slightly more damped motion compared to the head. The gluteal area shows the lowest angular acceleration values, mostly within the ±10–15 deg/s^2^ range, further confirming its role as a stabilizing segment with limited rotational freedom.

Overall, these results corroborate the findings from the angular displacement and linear acceleration analyses: during front-to-back rocking, the mechanical energy is most intense at the upper extremities (head), gradually diminishing toward the base (gluteal area). The high angular velocities and accelerations measured at the head, especially in the sagittal plane, emphasize the need for caution when designing or using infant rockers. Although the motion at the base may appear gentle, the inertial amplification at the head due to its distance from the pivot point can result in significantly larger and more abrupt rotational loads. For real infants, this could have implications for head and neck biomechanics, particularly during repeated or prolonged rocking. These data provide valuable quantitative evidence that can inform ergonomic design, safety guidelines, and potential thresholds for safe angular exposure in early infancy.

During gentle rocking motions, the acceleration forces experienced by different regions of an infant’s body typically range between approximately 0.2 and 1.7 G (Figure 4). Specifically, the head is subjected to forces varying from about 0.6 to 1.6 G, the abdomen between 0.2 and 1.7 G, and the pelvic region around 1.5 to 1.6 G. These values represent mild dynamic stimuli that are close to or slightly above normal gravitational force (1 G) and reflect the natural mechanical environment during soothing movements. At 2 to 3 months of age, an infant’s brain is undergoing rapid growth characterized by high neuroplasticity and ongoing synaptogenesis. The brain tissue at this stage is soft and well-cushioned by cerebrospinal fluid, providing some protection against mechanical stress. The range of acceleration forces described here is unlikely to pose any risk of injury; rather, such gentle stimulation may play a role in sensory integration and vestibular system development. Excessive or abrupt forces would be necessary to cause damage, which these measured values do not approach.

### 3.2. Wavelet Analysis

Figure 5 shows the results of a continuous wavelet transform (CWT) applied to the acceleration signals collected from three locations on the infant dummy: the head (forehead), abdomen, and gluteal region. The time window selected for analysis spans from 3 to 15 s of the experiment, capturing the segment of motion with the clearest oscillatory behavior before significant damping set in.

In the head wavelet transform, we observe a distinct band of activity between approximately 8 and 13 Hz, with intermittent bursts extending beyond 15 Hz. These higher-frequency components—more prominent in the head than in the other regions—suggest that the head is exposed to more abrupt, higher-frequency oscillations during the rocking motion. This is likely a result of the head’s increased mobility and distance from the point of mechanical excitation at the base. The head acts as the most dynamically sensitive segment, amplifying energy transmitted through the torso, which could be of biomechanical concern in the context of infant safety.

In contrast, the abdominal wavelet transform shows a slightly narrower frequency band of dominant activity, mainly between 8 and 12 Hz. The energy distribution is more uniform and less fragmented than in the head, reflecting smoother and less abrupt movement. The abdominal region appears to act as a transitional zone—receiving and partially attenuating the energy propagated upward from the base. This damping effect is consistent with what was observed in the time-domain acceleration plots.

The gluteal wavelet transform presents the most stable and narrowband signal, with energy concentrated mainly between 6 and 10 Hz. There is minimal high-frequency activity above 13 Hz, which suggests that the gluteal region undergoes relatively slow, controlled oscillations. This behavior is expected given its position in direct contact with the rocker’s seat, making it the mechanical origin of the motion. Notably, the gluteal signal acts almost like a baseline, providing insight into the characteristics of the input excitation transmitted to the upper body.

Taken together, the wavelet analysis supports a consistent pattern observed across previous kinematic metrics: the amplitude and frequency content of motion increase as we move from the gluteal base upward toward the head. This vertical amplification effect—particularly in the head—raises important considerations for the biomechanical safety of passive rocking in infants. While the motion may appear smooth at the base, wavelet-based frequency analysis reveals that higher frequency oscillations do reach the head, which may contribute to increased mechanical loading of the cervical spine and developing brain structures in real infants.

Figure 6 illustrates the time–frequency transfer function magnitude between body segments, offering insight into how motion energy propagates upward from the gluteal base through the abdomen to the head. Each subplot represents the dynamic relationship between two sensor locations, enabling an assessment of how motion at the base translates into movement further along the body chain, both across time and frequency ranges. Color represents the magnitude of frequency-specific coupling over time, revealing how motion energy propagates upward through the body segments. Notably, a concentrated energy transfer in the 10–12 Hz band is observed from the abdomen to the head, indicating strong dynamic coupling at these frequencies.

The first plot (Gluteal → Abdominal) shows a relatively low and diffuse transfer magnitude across the frequency range. There are no sharply localized peaks, which suggests that the energy transmitted from the base to the abdominal region is more uniformly spread out and possibly damped. This aligns with earlier observations from the wavelet and time-domain analyses, where the abdominal sensor behaved as a transitional filter, smoothing the input signal from the rocker. The gluteal-to-abdomen coupling appears relatively weak, especially in the higher frequencies (above 10 Hz), indicating that the abdominal region absorbs or buffers much of the input motion.

In contrast, the second plot (Abdominal → Head) shows a much stronger and more concentrated transfer function, particularly around the 10–12 Hz range between 10 and 20 s. This indicates a clear amplification of energy from the abdominal segment to the head, especially in that frequency band. The presence of a localized, high-magnitude region suggests a strong mechanical coupling and possibly a resonant-like transmission, where oscillations in the torso efficiently transfer into the head. This is consistent with the earlier findings of increased angular velocity and acceleration in the head, as well as higher frequency content in its wavelet transform. The head, as the least constrained and most distal segment, reacts strongly to motion initiated lower in the body—especially when that motion is modulated through the abdomen.

The third plot (Gluteal → Head) shows a more diffuse pattern, but still with some elevated transfer magnitude around 10–12 Hz. While the coupling is not as sharply defined as in the abdominal-to-head case, this still suggests that certain frequency components originating at the rocker base can propagate all the way to the head, even if partially filtered through the abdomen. The less defined shape and lower peak magnitude may reflect both damping effects and phase differences introduced by intermediate body segments.

Altogether, these transfer function results support the idea of progressive amplification of motion energy as it travels from the lower to upper body. Although the initial input at the gluteal level is relatively gentle and broadly distributed in frequency, the motion becomes more concentrated and potentially resonant by the time it reaches the head. This phenomenon underscores the importance of considering frequency-dependent transmission dynamics when evaluating the safety and biomechanical impact of infant rockers. What may appear as a low-frequency, low-intensity input at the base can lead to non-trivial motion loads at the head, particularly in vulnerable frequency bands.

## 4. Discussion

The present study offers detailed insights into the dynamic response of an infant doll subjected to passive anterior–posterior rocking in a common infant rocker, focusing on segmental motion captured via IMUs. These results align with earlier segmental motion research using wearable sensors in infants or children [24,25] which underscored variability across anatomical regions. While previous studies have examined segmental motion patterns and head kinematics in real infants during spontaneous movements or daily caregiving activities [24,25], no published research to date has directly quantified frequency-dependent motion transmission in commercial infant rockers. This absence of direct data highlights the exploratory and original nature of the present work, while the current findings provide a biomechanical reference framework for future in vivo validation studies.

Our kinematic analysis revealed that pitch rotation was the dominant motion across all segments, which is expected given the uniaxial rocking motion. The head exhibited the largest angular displacements, reaching initial pitch angles close to −60 degrees, followed by a damped oscillatory pattern indicative of a pendulum-like free response. This suggests that the head, due to its elevated position and relatively unconstrained mounting on the doll, undergoes more significant rotational excursions compared to the more stabilized torso and base segments. These observations align well with biomechanical expectations, as the head is mechanically the most distal point from the pivot, where inertial amplification can occur.

The abdominal segment, situated centrally and in contact with the rocker seat, showed smaller but still distinct pitch oscillations. The reduced amplitude likely reflects both mechanical damping from contact forces and internal structural constraints within the doll. The gluteal segment, fixed directly to the rocker base, exhibited the smallest angular displacements, consistent with its role as the motion origin transmitting energy upward rather than undergoing independent motion. The low variability in roll and yaw components further supports the predominantly sagittal nature of the rocking motion.

Linear acceleration data complemented these angular findings, reinforcing the top-down gradient of mechanical loading. The head experienced the highest peak accelerations—up to approximately 13 m/s^2^—shortly after rocking onset, with gradual decay over time. This pattern suggests that the head is more sensitive to rapid velocity changes and inertial forces, which could have important implications for infant safety, particularly given the vulnerability of the cervical spine and developing brain structures at early ages. The abdominal region exhibited moderate accelerations, while the gluteal segment showed the least dynamic loading, emphasizing its role as the stabilizing base. Similar magnitudes and hierarchical patterns of accelerations in infant head kinematics during daily activities were reported in biomechanical studies using surrogate models (e.g., shaking vs. ADLs) [26].

Importantly, angular velocity and acceleration analyses added further granularity by revealing the temporal and frequency characteristics of motion at each body segment. The sinusoidal, damped nature of angular velocity signals in the sagittal plane for the head and abdomen indicates a harmonic response to rocking. The sharper angular acceleration peaks seen in the head reflect rapid reversals of motion direction, which may translate to substantial mechanical stresses on the neck region in actual infants. This is consistent with concerns raised in pediatric biomechanics, where the immature musculoskeletal system may be less capable of safely dissipating such dynamic loads [27,28,29,30].

Wavelet-based time–frequency analysis provided additional depth by uncovering frequency-specific motion patterns. The head displayed higher frequency components up to and beyond 15 Hz, unlike the abdomen and gluteal segments, where dominant frequencies were lower and more narrowly concentrated. This suggests that the head not only experiences larger amplitude motions but also more abrupt, higher frequency oscillations, likely due to mechanical resonance phenomena and inertial amplification [18]. The abdominal region’s intermediate frequency range and more uniform energy distribution support its function as a transitional damping zone, while the gluteal region’s narrowband, low-frequency oscillations reflect its role as the input source.

The transfer function analysis between segments further elucidated the dynamics of energy propagation. The relatively diffuse, low-magnitude coupling between the gluteal and abdominal segments indicates substantial damping and energy dispersion early in the transmission chain. In contrast, the strong and focused transfer between abdomen and head in the 10–12 Hz band points to a resonant amplification effect, where motion energy concentrates and intensifies as it reaches the uppermost segment. This resonance-like behavior may pose biomechanical risks, as frequency-specific loading can disproportionately affect vulnerable tissues such as the developing brain and cervical spine [7,17].

Collectively, these findings underscore the complexity of passive rocking dynamics and highlight the importance of considering segmental biomechanics and frequency-dependent transmission when evaluating infant rocker designs. While the base motion may appear gentle and low frequency, inertial and resonant effects can amplify mechanical loads at the head. This has direct relevance for ergonomic and safety guidelines, suggesting that simply limiting rocking amplitude or frequency at the base may not be sufficient to ensure safe exposure levels at the head and neck.

From a developmental perspective, the measured acceleration magnitudes—ranging mostly between 0.2 and 1.7 G—are within the range of mild mechanical stimuli that infants might naturally encounter and may support vestibular system development without causing harm. However, the presence of higher frequency components and the potential for resonant amplification suggest caution, particularly for medically fragile infants or those with compromised neck muscle control. Future studies involving live subjects, alongside refined biomechanical modeling, will be essential to validate these findings and translate them into practical safety thresholds.

In conclusion, this segmental IMU analysis sheds light on the nuanced dynamics of infant rocking motion, emphasizing the critical role of frequency-dependent mechanical energy transmission and amplification. These insights pave the way for improved rocker design and usage recommendations aimed at minimizing potentially harmful inertial loads while preserving the soothing benefits of gentle rocking.

Clinically, the observed amplification of motion toward the head could be relevant for cervical spine loading and neuromotor control in early infancy. Infants, particularly those born preterm or with neurological impairments, may have reduced capacity to dampen oscillatory motion, potentially altering mechanical stress distribution. While our study focused on biomechanical aspects, these findings may provide useful context for future clinical and ergonomic evaluations of infant rockers.

## 5. Limitations

Limitations of this study should be acknowledged. First, the use of an infant doll, while useful for controlled biomechanical measurement, cannot fully replicate the complex musculoskeletal and neuromuscular properties of a living infant, including active muscle control and reflexive stabilization. The doll’s rigid body segments and simplified anatomy may influence damping and transmission characteristics differently from real infants. Moreover, the passive mannequin model cannot replicate active muscle tone, reflexive head stabilization, or spontaneous infant movement, all of which could significantly affect damping behavior, frequency response, and inter-segmental variability in real infants. Second, the study focuses on a single rocker model and rocking frequency range, which limits generalizability to other rocker designs or more varied motion patterns. Third, the IMU placement and measurement approach, although precise, may have minor alignment errors that affect absolute angular values. Finally, the study does not account for long-term effects or repeated rocking exposures, which are relevant for understanding cumulative biomechanical loading.

Future research should aim to extend these findings by incorporating biomechanical models of infant anatomy with active muscle properties and reflexes to better simulate physiological responses. Studies involving real infants, using non-invasive motion capture and physiological monitoring, would provide valuable validation and insights into individual variability. Additionally, exploring a broader range of rocking devices, motion amplitudes, frequencies, and durations will help define safe exposure thresholds more accurately. Investigation of the effects of different seating positions, support systems, and rocker designs on segmental dynamics would be valuable for optimizing ergonomic safety. Finally, integrating these biomechanical data with developmental neuroscience could clarify the role of passive rocking in sensory integration and motor development while ensuring infant safety.

Together, these avenues will help translate quantitative biomechanical knowledge into improved product standards and evidence-based recommendations, ultimately enhancing the safety and well-being of infants during passive rocking.

## 6. Conclusions

This study provides a comprehensive motion analysis of an infant rocker using IMUs, revealing how mechanical energy from passive anterior–posterior rocking propagates through the body segments of an infant doll. The data clearly demonstrate that while the base of the rocker exhibits gentle, low-frequency oscillations, the head experiences amplified angular displacements, velocities, and accelerations, especially in the sagittal plane. These findings highlight the potential biomechanical implications for infant safety, emphasizing the need to consider inertial and resonant effects in rocker design and usage guidelines.

## Figures and Tables

**Figure 1 jcm-14-08301-f001:**
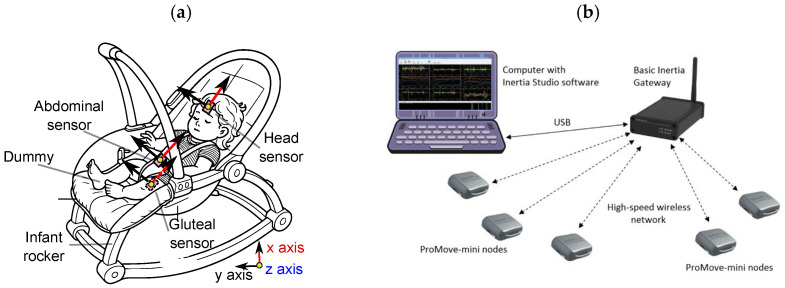
Baby rocker with dummy and IMU sensor placement showing sensor orientations (**a**) Experimental system setup [21]. (**b**) Preliminary processing after data collection.

**Figure 2 jcm-14-08301-f002:**
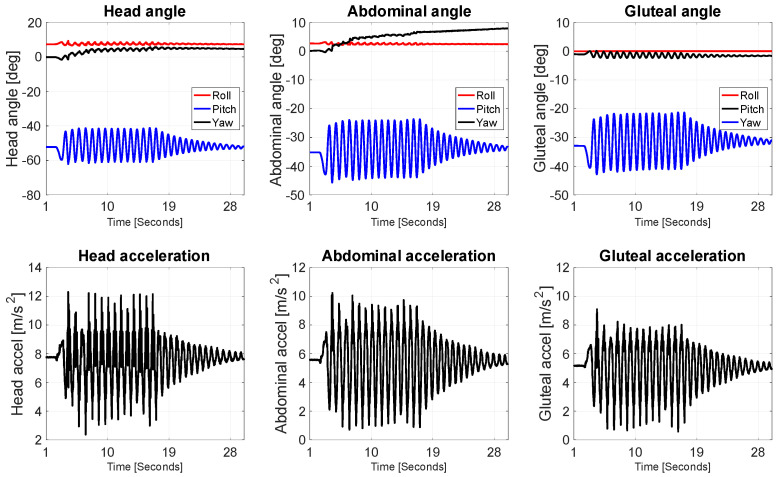
Measured angles and linear acceleration (head, abdominal, gluteal sensors).

**Figure 3 jcm-14-08301-f003:**
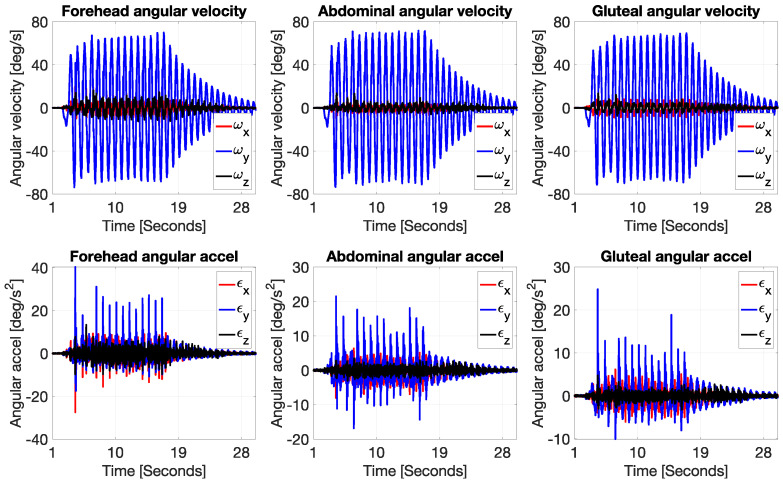
Measured angular velocity and acceleration (head, abdominal, gluteal sensors).

**Figure 4 jcm-14-08301-f004:**
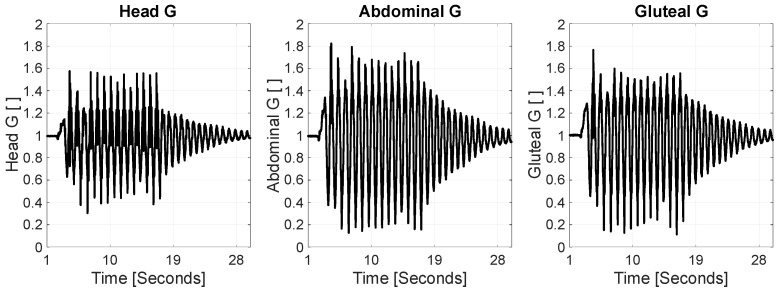
G-force at dummy’s head, abdominal, and gluteal sensors.

**Figure 5 jcm-14-08301-f005:**
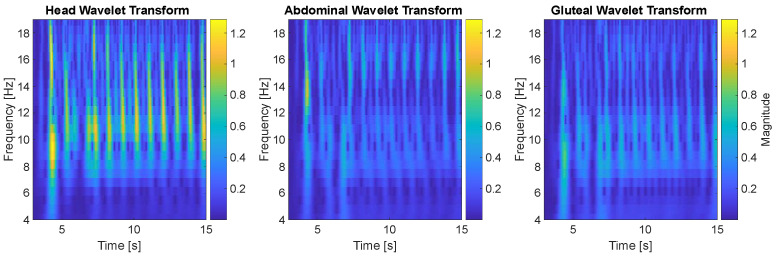
Continuous Wavelet Transform of Head, Abdominal and Gluteal acceleration signals.

**Figure 6 jcm-14-08301-f006:**
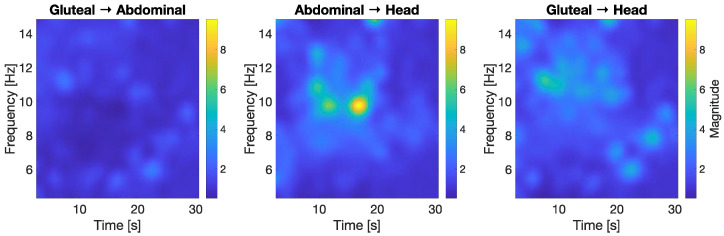
Time–frequency transfer function magnitude between gluteal, abdominal, and head acceleration signals.

## Data Availability

The data presented in this study are available on request from the corresponding author (included in manuscript). The data are not publicly available due to file size and data management limitations.

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
