# Peer review of "Frequency-Dependent Amplification of Head Motion in Infant Rockers: A Segmental IMU-Based Signal Analysis"

_jcm, 2025, doi:10.3390/jcm14238301_

Round 1
Reviewer 1 Report
Comments and Suggestions for Authors
The study provides crucial insights into the biomechanics of passive rocking, confirming that the head experiences significant dynamic amplification compared to lower body segments. The findings are clearly presented, showing that the head exhibited the highest angular displacements and linear acceleration peaks (up to approximately 13 m/s²). The wavelet and transfer function analyses revealed a concentrated energy amplification in the 10–12 Hz frequency band between the abdomen and the head—an important result suggesting a resonance or strong coupling effect that is highly relevant for the ergonomic safety of infant rockers. The segmental behavior is also well characterized: the gluteal region acts as the mechanical input point (static base), the abdomen serves as a damping or transitional filter, and the head—the most distal and least constrained segment—shows the most dynamic oscillatory response. Methodologically, the application of the Continuous Wavelet Transform (CWT), particularly with the Morlet wavelet, is well-justified for analyzing the non-stationary, quasi-periodic motion signals generated during passive rocking.
The main limitation of the study lies in the use of a rigid mannequin model, which lacks active muscular control and the neuromuscular properties of a real infant—factors that may influence damping and energy transmission. Future research could benefit from incorporating biomechanical models that simulate active muscular and reflex mechanisms, as well as non-invasive validation studies with real infants to better establish practical safety thresholds.
Overall, this is a highly relevant and timely study that contributes to understanding the mechanical implications of passive rocking on infant safety. If feasible, adding a brief statistical analysis or quantitative comparison (e.g., across frequency bands or body segments) could further strengthen the robustness of the findings and enhance their interpretive value.
Author Response
Dear Reviewer,
we would like to sincerely thank you for the time and effort devoted to evaluating our manuscript and for the valuable comments and suggestions provided. Below, in the attached file, we address each of the Reviewer’s remarks in detail. Responses are marked in red color.
Yours Sincerely
Prof. Sebastian Glowinski

Reviewer 2 Report
Comments and Suggestions for Authors
The manuscript presents an interesting and timely analysis of motion transmission in infant rockers using IMU-based segmental measurements. This is a very interesting topic, overall, the study has merit, but several aspects would benefit from deeper elaboration with minor revisions:
1) A first point concerns the clinical perspective, which at present is only briefly addressed. The manuscript convincingly shows that oscillatory input at the rocker base can undergo frequency-dependent amplification as it propagates upward to the head. However, the discussion does not fully explain why this phenomenon might be clinically significant, nor what it could mean for cervical spine loading, head–neck stabilisation, or developing neural tissues. Given the vulnerability of infants in their first months of life, it would be very helpful to expand this section and clarify how the observed mechanical patterns might – cautiously – translate into potential clinical considerations. Is there the possibility to have differences between newborns born preterm and those at term? Or maybe in infants with neurological sequelae?
2) I would encourage the authors to expand the comparison with literature data, especially regarding studies that have attempted to measure head or torso dynamics in real infants in everyday settings. If no studies exist that directly examine motion transmission in commercial rockers, that absence should be clearly stated, as it would reinforce the originality and exploratory nature of this work. Conversely, if partial or indirect evidence exists, it would be useful to compare the present findings with those previous data.
3) The authors already acknowledge that the study relies on a single experimental trial performed on a non-biological infant mannequin, which is an important limitation to be explained in the limitation section, because a passive manikin cannot replicate active neuromuscular responses, reflexive head stabilization, or spontaneous movement, all of which would likely influence damping, frequency response, and real-world variability.
4) Please note that there are still some parts taken from the journal template that need to be removed (abstract, line 17).
Author Response
Dear Reviewer,
We would like to sincerely thank you for the time and effort devoted to evaluating our manuscript and for the valuable comments and suggestions provided. Below, in the attached file we address each of the Reviewer’s remarks in detail.
Changes in the manuscript are in red.
Yours Sincerely
Prof. Sebastian Glowinski
